# Effect of 4-Week Consumption of Soy Kori-tofu on Cardiometabolic Health Markers: A Double-Blind Randomized Controlled Cross-Over Trial in Adults with Mildly Elevated Cholesterol Levels

**DOI:** 10.3390/nu15010049

**Published:** 2022-12-22

**Authors:** Maartje van den Belt, Sandra van der Haar, Els Oosterink, Tom van Loenhout, Takahiro Ishiguro, Diederik Esser

**Affiliations:** 1Wageningen Food and Biobased Research, Wageningen University & Research, 6708 WG Wageningen, The Netherlands; 2Department of Cardiology, Ziekenhuis Gelderse Vallei, 6710 HN Ede, The Netherlands; 3Food Research Laboratory, Asahimatsu Foods Company Limited, Nagano 399-2561, Japan

**Keywords:** Kori-tofu protein, soy protein, total cholesterol, LDL cholesterol, cardiometabolic risk factors, glucose metabolism, blood pressure

## Abstract

Kori-tofu is a frozen soy tofu, and soy consumption is associated with positive effects on cardiometabolic health markers. We aimed to assess the potential of Kori-tofu to improve cardiometabolic health outcomes in humans by repetitive daily consumption. In a double-blind randomized controlled cross-over trial, 45 subjects aged 40–70 years with (mildly) elevated cholesterol levels, received a four week Kori-tofu intervention or whey protein control intervention with a four week wash-out period in between. Cardiometabolic biomarkers were measured before and after both interventions. A significant decrease in total, low-density lipids (LDL), and high-density lipids (HDL) cholesterol, Hemoglobin A1c (HbA1c), fructosamine and systolic blood pressure was observed within the Kori-tofu intervention. However, many of these findings were also observed in the control intervention. Only adiponectin changes were different between treatments but did not change significantly within interventions. Improvements in cardiometabolic markers within the Kori-tofu intervention point toward potential beneficial health effects. Due to the lack of significant effects as compared to control, there is, however, currently no substantiating evidence to claim that Kori-tofu has beneficial effects on cardiometabolic health.

## 1. Introduction

Elevated cholesterol levels are one of the most important modifiable metabolic risk factors for developing cardiovascular diseases (CVD) [1]. In addition, several other cardiometabolic markers such as impaired glucose tolerance and high blood pressure contribute to the risk of developing both CVD and type 2 diabetes [2,3,4]. Therefore, treating unfavourable metabolic profiles at an early stage is essential in reversing disease risk. Even though age and genetics can increase the risk of developing metabolic risk factors, more often it is the result of unhealthy lifestyle choices [5]. Lifestyle interventions, including diet and exercise, are preferred as the first-line treatment, before starting drug treatment [6]. In this regard, nutrition can be a powerful tool to improve metabolic health. This raises the interest in and need for foods with beneficial health properties, for example, foods being naturally rich in bioactive components.

Soy protein is known for its positive effect on cardiometabolic health markers. A recent meta-analysis of 46 studies demonstrated that soy protein consumption at a median dose of 25 g per day for 6 weeks decreased both low-density lipids (LDL) cholesterol and triglyceride levels [7]. Another meta-analysis of 18 studies demonstrated the beneficial effects of soy milk consumption on several cardiometabolic health markers, such as blood pressure (BP) and LDL and total cholesterol levels [8]. Next to beneficial effects on lipid metabolism, soy protein consumption has also been studied in relation to improving glucose metabolism and BP. A recent meta-analysis of 12 trials showed that soy protein and isoflavine consumption may cause a reduction, although not significant, in fasting glucose, insulin, Hemoglobin A1c (HbA1c), and Homeostatic Model Assessment of Insulin Resistance (HOMA-IR) in diabetic patients [9]. Another meta-analysis including diabetic and healthy participants, as well as obese, overweight, hypercholesterolemic, and hypertensive participants, showed no beneficial effects of soy protein consumption on glucose metabolism [10]. Both systolic and diastolic blood pressure was positively affected by soy protein intake as compared to a control diet according to another meta-analysis. These reductions were, however, related to pre-treatment BP levels of subjects and the type of control diet used as comparison [11]. All of the above mentioned markers are therefore interesting to study further in the context of improving cardiometabolic profiles by foods containing high amounts of soy protein.

Tofu, also known as bean curd, is a food product that is naturally rich in soy protein. It is produced by coagulating soy milk and then pressing the resulting curds into solid white blocks of varying softness. In East Asian and Southeast Asian cuisines, tofu is a traditional food and it has been consumed in China for over 2000 years. An example of a specific type of tofu is ‘Kori-tofu’, which literally means frozen tofu. Kori-tofu can be used as an ingredient to replace wheat flour, comparable to regular soy flour. Therefore, it can be added to a wide range of foods, such as snacks and bakery products. The production process involves high-pressure press treatment, freezing, low-temperature ageing, and drying. This process affects the texture of the product by decreasing the amount of moisture to 8%, resulting in a sponge-like texture absorbing water and flavour during the cooking process. The production process also leads to the formation of a larger high molecular weight fraction (HMF) content of the soy proteins [12]. Kori-tofu protein is therefore different from regular soy protein. Biochemical analysis demonstrated that the HMF content of Kori-tofu protein was significantly higher than that of soy protein isolate [13]. Scientific data from rat experiments demonstrate that Kori-tofu may decrease serum LDL cholesterol and triacylglycerol even further than a soy protein isolate [13]. The current hypothesis is that the HMF of Kori-tofu can bind to bile acids in the intestine, hence preventing bile acid resorption and thereby affecting cholesterol absorption in the intestine [14,15]. It has been shown, also in rats, that the HMF content is most likely responsible for this increase in excretion of bile acid, and other lipids in faeces [16]. Recent mechanistic studies in both animals and humans have demonstrated that bile acids bind to their endogenous receptors, including Farnesoid X Receptor (FXR) and G protein-coupled bile acid receptor 1 (GPBAR1; TGR5), which results in the secretion of gastrointestinal hormones. Hormones such as Ghrelin, Cholecystokinine (CCK), Glucagon-like peptide 1 (GLP-1), Peptide YY (PYY), and insulin thereby control various biological processes, including cholesterol-bile acid metabolism, glucose-lipid metabolism, and energy metabolism [15,17].

Although beneficial metabolic health effects of soy protein are described in the literature, in vivo human evidence of beneficial health effects of Kori-tofu is currently lacking. Therefore, the current study aims to assess the effect of repetitive daily consumption of Kori-tofu on blood total and LDL cholesterol levels as compared to a control intervention with whey protein. Furthermore, we are interested in the effects on other markers of lipid and glucose metabolism, and cardiometabolic outcome measures.

## 2. Materials and Methods

### 2.1. Participants

All participants provided informed consent for participation in the study. The study was conducted in accordance with the Declaration of Helsinki and the protocol was approved by the Ethics Committee of Utrecht University (Utrecht, The Netherlands) (file number NL75320.081.20). The study was registered at clinicaltrials.gov (NCT04896619). Study execution was between September and December 2021 at Wageningen University & Research in the Netherlands.

The study population consisted of men and women, aged 40–70 years at the time of recruitment with (mildly) elevated levels of cholesterol. Participants were recruited via advertisements on social media and in newspapers, and via a database of Wageningen University & Research, between June 2021 and September 2021. Inclusion criteria were: BMI between 18.5–35 kg/m^2^ and total cholesterol levels > 5 mmol/L or LDL cholesterol levels > 3 mmol/L. The main exclusion criteria were: use of medication or supplements that could influence the study outcome, total cholesterol levels > 7 mmol/L or LDL cholesterol levels > 5 mmol/L, reported slimming diets or weight loss or weight gain of >5 kg in the month before screening, and current smokers. Eligibility was assessed using a screening visit, where a questionnaire was administered to check medical history. Furthermore, fasting cholesterol levels and body mass index (BMI) were measured.

### 2.2. Study Design and Procedures

This study had a double-blind randomized controlled cross-over design. All participants received two 4-week interventions (Kori-tofu protein or whey protein) with a washout period of 4 weeks in between. At the beginning and end of each intervention period, participants visited the research unit for a test morning (T = 0, T = 4, T = 8 and T = 12 weeks), baseline samples were collected at T = 0 and T = 8 weeks and endline samples at T = 4 and T = 12 weeks. Participants consumed a standardised meal the night before the test day and came to the facility after an overnight fast. Each test morning, body weight and blood pressure were measured and blood was collected. At the end of each test morning, participants received breakfast on location. During the entire trial, participants were instructed to maintain their normal routines regarding their diet and exercise patterns and were instructed to avoid weight gain or weight loss. Body weight was assessed at each study visit and monitored throughout the study (T = 0, T = 2, T = 4, T = 8, T = 10 and T = 12 weeks).

### 2.3. Intervention Products

The Kori-tofu and whey protein isolates were mixed in wheat bread. Bread was chosen as an intervention product since this is the most consumed product for breakfast and lunch in the Netherlands. The intervention meals consisted of 4 daily slices or buns of wheat bread (2 at breakfast, and 2 at lunch), containing a total of 34 g of Kori-tofu per day (51.5% proteins, 34.5% lipids, and 3% carbohydrates) or control which was matched with the same amount of whey protein and sunflower oil. To ensure sufficient variation, from Monday to Thursday participants were instructed to eat slices of bread and on weekend days (Friday to Sunday) they received buns. Each slice or bun of bread contained approximately 8.6 g of Kori-tofu. In Table 1 the energy and macronutrient content of both intervention products are displayed.

Participants could choose their bread toppings. They were however instructed to choose bread toppings that they usually eat, in the same amount. Participants completed a daily diary to report compliance and consumption of their meals. Halfway through each intervention period, an in-between visit was organised. During these visits and at the end of each intervention, product compliance was checked by counting left-over slices and buns. Furthermore, body weight was measured and new test products were provided for the next two weeks.

### 2.4. Study Measures

#### 2.4.1. Analysis of Lipid and Glucose Metabolism

Total cholesterol, high-density lipids (HDL) cholesterol, triglycerides, plasma insulin, HbA1c, fructosamine, and glucose levels were determined in all collected blood samples. Blood samples were immediately processed after each collection and plasma samples were stored at −80 °C till analyses at Ziekenhuis Gelderse Vallei, Ede, the Netherlands. LDL cholesterol levels were determined based on the Friedewald calculation [18]. HOMA-IR scores were calculated according to the following formula: fasting insulin × fasting glucose/22.5 [19].

#### 2.4.2. Blood Pressure

Blood pressure (systolic and diastolic) was measured with an automatic blood pressure monitor (Omron Healthcare Europe, Hoofddorp, The Netherlands, HEM-907). To ensure reliable measurements, participants had to sit down and rest for at least 10 min before the measurement started. During each study visit, blood pressure was measured at least 3 times. In case of deviations larger than 10 mmHg between the second and third measurements, a fourth measurement was performed. For the calculation of mean values, the first measurement was discarded.

#### 2.4.3. Adiponectin and Leptin

Adiponectin and leptin levels were measured in all collected blood samples. Blood samples were immediately processed after collection and serum samples were stored at −80 °C till analysis. Serum samples were thawed on ice, diluted, and measured with commercially available ELISA kits (Human ADIPQ ELISA kit and Human Leptin ELISA kit, Merck life Sciences, Bengaluru, India). Assays were performed according to the provided protocol with 10% of the samples measured in duplicate. A linear regression standard curve was generated using GraphPad Prism (version 5) to interpolate sample values.

### 2.5. Sample Size and Randomization

A sample size calculation was performed with an 80% chance of detecting a mean reduction of 1.7% in total cholesterol, at a two-sided significance level of 0.5, with an assumed standard deviation (SD) of 0.13 [20]. This resulted in a sample size of 39 participants, which are needed to be able to demonstrate a significant difference between the two interventions. Taking into account potential dropouts, the number of participants was increased to 48. The order in which participants received both interventions [AB or BA] was randomized by using block randomization. The variables gender, age, and total cholesterol levels were stratified among the treatment orders.

### 2.6. Statistical Analysis

All values are expressed as mean ± SD unless stated otherwise. For comparison analyses between the two different interventions, delta values between week 4 and baseline were calculated and analysed using paired sample *T*-tests. To evaluate the change within an intervention, baseline values were compared to end values, with a paired sample *T*-test. To check if the wash-out period of 4 weeks was sufficient enough, the period effect was calculated by comparing the baseline values of period 1 and period 2. These were also analysed with a paired sample *T*-test. A *p*-value ≤ 0.05 was considered statistically significant. Since dietary interventions usually elicit small effects, no multiple testing corrections were applied. All statistical analyses were performed using IBM SPSS statistics version 25.0.

## 3. Results

### 3.1. Participants, Baseline Characteristics and Compliance

Figure 1 shows a flowchart diagram of the participant selection and randomization process. Of the 48 randomized participants, 2 participants withdrew during the study because of personal reasons, and 1 participant was not able to complete the study because of a COVID-19 infection. A total of 45 participants (19 males and 26 females) completed the study. Their mean age was 59 years, ranging from 44–69 years and their mean BMI was 25.4 kg/m^2^, ranging from 20.3–34.2 kg/m^2^. Baseline total and LDL cholesterol levels were on average 5.7 mmol/L and 3.4 mmol/L, respectively. The average compliance of consumption of the intervention products was above 99% for both interventions based on the counting of left-over slices and buns. In both interventions, a significant increase in body weight was measured (Kori-tofu: 0.4 kg, Control: 0.3 kg, *p* < 0.01 within both interventions). The increase in body weight was, however, not significantly different between the two interventions (*p* > 0.05).

### 3.2. Cardiometabolic Health Markers

Table 2 (and Appendix A Figure A1, Figure A2 and Figure A3) shows the cholesterol and triglycerides values at baseline and the change after 4 weeks intervention within the Kori-tofu or control intervention. Within the Kori-tofu intervention, total cholesterol levels decreased significantly with a reduction of 0.14 mmol/L, which corresponds to a reduction of 2.5% from baseline. Total cholesterol levels did not change significantly within the control intervention. The change after the Kori-tofu intervention was not significant when compared to control. LDL cholesterol levels decreased significantly within both the Kori-tofu and the control intervention. A significant LDL reduction of 0.27 mmol/L, which corresponds to a reduction of 7.3% from baseline, was observed in the Kori-tofu intervention and a significant LDL reduction of 0.23 mmol/L, corresponding to a reduction of 6.1% from baseline, was observed in the control intervention (*p* ≤ 0.01 for both). HDL cholesterol levels also significantly decreased within both interventions. A significant HDL reduction of 0.08 mmol/L was observed within the Kori-tofu intervention, corresponding to a reduction of 5.3% from baseline (*p* ≤ 0.01), a significant reduction of 0.05 mmol/L was also observed in the control intervention, corresponding to a reduction of 3.3% from baseline (*p* ≤ 0.01). A significant increase in triglyceride levels was observed within the control intervention when compared to baseline (*p* = 0.02), but not in the Kori-tofu intervention. None of the observed changes in total cholesterol, LDL, HDL, and triglycerides were significantly different between the Kori-tofu intervention and the control intervention.

Baseline LDL and total cholesterol levels between the first and second intervention period were compared, to evaluate whether there was any carry-over effect of the former intervention period. LDL and total cholesterol levels did not significantly differ at the start of the two intervention periods (*p* ≤ 0.05). To evaluate whether baseline cholesterol levels could partly explain the magnitude of response, additional subgroup analyses were performed. Participants with baseline values below or equal to the median (total cholesterol ≤ 5.5, LDL ≤ 3.7 mmol/L) were classified in the low baseline sub-group and participants with baseline values higher than the median in the high baseline sub-group. No significant difference in response was observed between the two subgroups for both total cholesterol (Kori-tofu: *p* = 0.88; control: *p* = 0.23) and for LDL cholesterol (Kori-tofu: *p* = 0.27; control: *p* = 0.27).

Table 3 (and Appendix B, Figure A4 and Figure A5) shows the baseline values and the changes in markers of glucose metabolism after 4 weeks intervention with Kori-tofu or control. A significant decrease of 6.1 μmol/L fructosamine, a 2.3% reduction from baseline, and 0.42 mmol/mol reduction in HbA1c, a 1.1% reduction from baseline, was observed within the Kori-tofu intervention. The changes observed in the control intervention, however, were not statistically significant. Glucose, insulin and HOMA-IR did not change within the interventions. Changes for glucose, insulin, fructosamine, and HOMA IR score were not significantly different between Kori-tofu and control intervention. However, the difference in response in HbA1c between the Kori-tofu and control intervention was almost significant (*p* = 0.06).

Table 4 (and Appendix C, Figure A6) shows blood pressure and heart rate values at baseline and the change after four weeks intervention with Kori-tofu or control. A significant decrease in systolic blood pressure was observed only within the Kori-tofu intervention. A significant increase in heart rate was observed only within the control intervention. The observed changes in blood pressure and heart rate were not different between the Kori-tofu intervention and the control intervention.

Table 5 shows leptin and adiponectin values at baseline and the change after four weeks intervention with Kori-tofu or control. A significant difference in change between the Kori-tofu and control intervention was observed for adiponectin levels (*p* = 0.03). The adiponectin levels were, however, not significantly affected within each intervention; a non-significant decrease in adiponectin was observed within the Kori-tofu intervention (*p* = 0.06) and a non-significant increase was observed in the control intervention (*p* = 0.13).

## 4. Discussion

This double-blind randomized controlled cross-over study showed that a 4-week intervention with Kori-tofu protein resulted in an improvement of the cardiometabolic profile as compared to baseline values. The within intervention comparison resulted in a significant decrease in total cholesterol, LDL cholesterol, HDL cholesterol, HbA1c, fructosamine and systolic blood pressure. These changes, however, did not significantly differ from the control intervention with whey protein. Changes in adiponectin levels were different between treatments but did not change significantly within the interventions.

Based on the lack of significant effects as compared to the control intervention, we do not have substantiating evidence to claim that Kori-tofu has beneficial effects on cardiometabolic health. However, although not statistically different from control, clear effects on cardiometabolic markers were observed within the Kori-tofu intervention, despite a high between-subject variation. These changes, as compared to baseline values, can provide valuable information on potential effect sizes and potential clinical meaningful changes. When looking at the within-intervention effect only, four weeks consumption of bread containing 34.5 g Kori-tofu per day resulted in a reduction of 0.14 mmol/L of total cholesterol, corresponding to a 2.5% reduction from baseline. LDL cholesterol was reduced by 0.27 mmol/L, a 7.3% reduction from baseline. A meta-analysis of 26 clinical trials evaluating the effect of the cholesterol lowering drug statins on LDL cholesterol and cardiovascular events, described a 22% reduction in major cardiovascular events for every 1.0 mmol/L reduction in LDL cholesterol [21]. Our observed reduction in cholesterol levels in the Kori-tofu intervention is in that respect relatively modest. Small effect sizes are common in nutritional interventions, but can certainly have a big impact in the long term and on a population level [22]. Our observed effect sizes are lower as compared to the effect of other bioactive foods associated with lowering cholesterol, such as plant sterols and stanols, which have an EFSA approved health claim and are proven to reduce total and LDL cholesterol levels [23]. A meta-analysis of 41 studies demonstrated that plant sterol or stanol consumption at a median dose of 2 g per day, during a median intervention period of 5 weeks, significantly reduced LDL cholesterol by 10% [24,25]. Another meta-analysis of 124 studies confirmed that the LDL-lowering effect of both plant sterols and stanols continues to increase up to intakes of around 3 g per day to an average lowering effect of 12% [26]. Most of these studies included subjects with on average higher LDL cholesterol levels compared to this study, which may partly explain the stronger relative reduction [24,25]. Next to the reduction in total and LDL cholesterol we also observed a reduction in HDL cholesterol, but these effects were relatively small.

Kori-tofu is derived from soy and has a high protein content. A recent meta-analysis containing data from 46 clinical trials reported that soy protein, at a median dose of 25 g per day and during a median intervention period of 6 weeks, resulted in a significant 3.2% reduction of LDL cholesterol. It must be noted that most of the studies in this meta-analysis that showed a reduction in LDL cholesterol were intervention studies with a duration of a minimum of 6–12 weeks. Furthermore, 20 of these studies included subjects with hypercholesterolemia [7]. Our Kori-tofu intervention contained a soy protein content of 17.5 g per day, which is somewhat lower than the median dose of 25 g per day described in this meta-analysis. However, the effects observed within the Kori-tofu intervention may be contributed to the soy protein content of the Kori-Tofu, rather than specifically the HMF content of Kori-tofu. To what extent the effect of Kori-tofu protein on cholesterol levels differs from regular soy protein cannot be estimated from our study, since we did not make this comparison. To study the potential effect of the HMF content of Kori-tofu a reference intervention containing soy protein should be included in follow-up studies.

Within intervention comparisons also showed that the Kori-tofu intervention lowered markers related to glucose homeostasis. The 0.4 mmol/mol reduction in HbA1c, equal to a 1.1% reduction from baseline, and the 6.1 μmol/L reduction in fructosamine, a 2.3% reduction from baseline, were significant within the Kori-tofu intervention and did not change within the control intervention. Still, between-subject variation in the response was high for these variables and when comparing both interventions, no significant difference was observed, although this was almost significant for HbA1c with a *p*-value of 0.06. HbA1c and fructosamine are both markers of glycaemic control and a reduction in these parameters are considered a beneficial metabolic effect. HbA1c reflects an average of blood glucose concentrations over 2–3 months [27,28]. Fructosamine reflects the average blood glucose concentrations over a shorter period, e.g., the past 1–3 weeks [28]. An observational study within 3642 participants showed that each 1% decrease in HbA1c was associated with a significant reduction of 21% in diabetes-related deaths, 14% in myocardial infarction and 37% in microvascular complications [29]. Previous studies investigating the effect of soy protein on HbA1c levels have found no effects [30]. This indicates that the reduction in HbA1c levels found in this study might be attributed specifically to the effect of Kori-tofu protein. In the current study, however, we did not compare Kori-tofu protein with soy protein, so this hypothesis should be studied further in more detail.

Furthermore, the decrease in systolic blood pressure within the Kori-tofu intervention also points towards a favourable effect on cardio-metabolic health, although a blood pressure lowering effect of 3 mmHg is relatively small and between-subject variation in response was high. Potassium levels of the Kori-tofu bread were slightly higher compared to the control bread which might explain the decrease in blood pressure after the Kori-tofu intervention [31,32]. However, blood pressure can be affected by numerous factors and this study was not designed to study the effects of interventions with different potassium contents. Still, this might be an interesting lead to study further in more detail.

Taken together, changes within the Kori-tofu intervention point towards an improved cardiometabolic profile, known to lower cardiovascular disease risk [33]. These effects were observed in a highly relevant study population with mildly elevated levels of cholesterol and therefore an unfavourable cardiometabolic profile. It also seems that the magnitude of response was not dependent on the participant’s baseline value, since participants with relatively high baseline values of total and LDL cholesterol responded similarly to those with lower baseline values of total and LDL cholesterol. This suggests, although highly speculative, that Kori-tofu can play an important role in prevention, and that it may exert its positive effect even at an early stage. Especially since this ingredient can be easily incorporated into various foods.

Although the positive effects on cardiometabolic profiles were not significantly different between the two interventions, some of these outcomes were significantly different within the Kori-tofu intervention and not within the control intervention. Effects seemed also more pronounced in the Kori-tofu intervention. The lack of significant effects between interventions and some improvements observed in both interventions can perhaps, at least partly, be attributed to changes in lifestyle during the study. Study participants were included based on their elevated levels of cholesterol. Receiving this kind of information may have changed their daily life activities, even though they were instructed not to. To what extent these aspects affected the results in both interventions is not known, but it emphasises the importance of performing a cross-over study with a control intervention as reference, to draw firm conclusions. LDL and total cholesterol levels did not significantly differ at the start of the two intervention periods. We, therefore, consider the wash-out period of four weeks sufficient, and do not think a carry-over effect from the first intervention period towards the second period would have had impact on the between-intervention comparisons.

Whey protein was chosen as control intervention to match the protein content of the Kori-tofu. Whey is a commonly used protein to match macronutrient content; it is easy to process in a final product because of its neutral flavour and solubility properties. Whey protein and high protein diets in general can, however, also have a beneficial effect on the cardiometabolic profile [34,35,36]. Previous studies have indicated that whey protein can reduce total and LDL cholesterol, triglycerides, HbA1c, and insulin levels [36]. Several potential mechanisms for the effect of whey protein on these cardiometabolic markers are described in literature [36]. For that reason, the choice of whey protein as a control in the current study is a point of discussion, since it may have exerted an effect on improving lipid metabolism markers. Compliance of the participants to the protocol was high and participants consumed nearly all slices of bread according to the study protocol. An unintended observed effect in our study was the small, but significant increase in body weight. This is most likely caused by the relatively high number of bread slices participants had to consume each day, four slices a day may be more than they were used to eating. Nevertheless, the increase in body weight was not different between the two interventions and despite this small increase, improvements in cardiometabolic profiles were observed. Another important factor for interpreting these results is that we did not correct for multiple testing in our statistical analyses. We decided not to apply such a correction because effect sizes in nutritional interventions are relatively small. More research is however needed to draw firm conclusions. Since this is a first human intervention trial with Kori-tofu protein, we wanted to identify and explore any potential benefit.

## 5. Conclusions

Observed improvements in cardiometabolic profiles after the Kori-tofu intervention did not significantly differ from control. We therefore found no clear substantiating evidence to claim that Kori-tofu has beneficial effects on cardiometabolic health. Still, within the Kori-tofu intervention we did observe positive effects on cardiometabolic health in participants with an impaired cardiometabolic profile. Especially the change in measures related to glycaemic control can be considered interesting in this respect.

## Figures and Tables

**Figure 1 nutrients-15-00049-f001:**
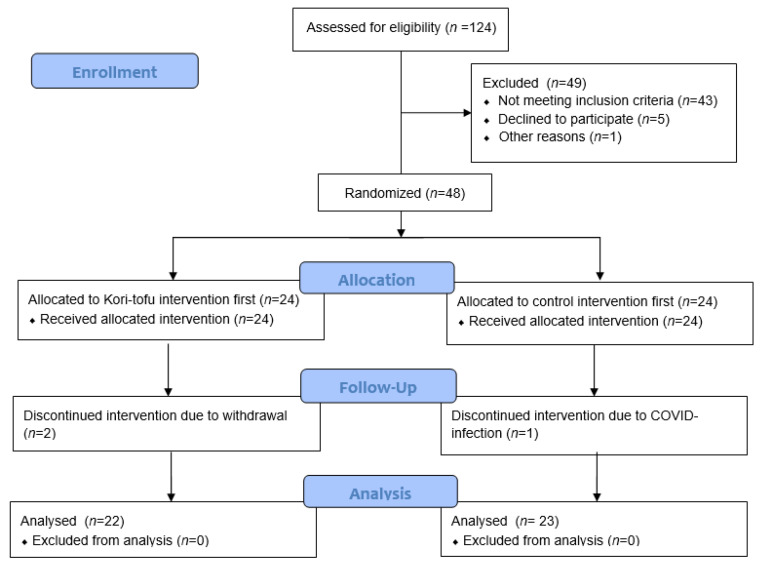
Flow-chart diagram of the participant selection and randomization process.

**Table 1 nutrients-15-00049-t001:** Energy and macronutrient content of 4 slices or buns of bread (Kori-tofu and whey protein), based on raw weight.

	Kori-tofu Bread (Kori-tofu Intervention)	Whey Protein Bread (Control Intervention)
Amount 4 slices/buns (g)	228	160
Energy (kcal)	472	472
Energy (kjoule)	1988	1984
Protein total (g)	27.6	27.6
Fat total (g)	13.2	13.2
Carbohydrates total (g)	57.2	57.2
Potassium (mg)	684	416

**Table 2 nutrients-15-00049-t002:** Baseline values and changes in total cholesterol, low-density lipids (LDL) cholesterol, high-density lipids (HDL) cholesterol and triglyceride levels after 4 weeks Kori-tofu or control intervention ^1^.

	Kori-tofu Intervention			Control Intervention			Between Kori-tofu and Control
	Baseline	Change ^2^	*p*-Value	Baseline	Change ^2^	*p*-Value	*p*-Value
Cholesterol (mmol/L)	5.7 ± 0.6	−0.14 ± 0.4	**0.02**	5.7 ± 0.7	−0.05 ± 0.5	0.44	0.34
LDL chol. (mmol/L)	3.7 ± 0.7	−0.27 ± 0.4	**<0.01**	3.8 ± 0.8	−0.23 ± 0.4	**<0.01**	0.71
HDL chol. (mmol/L)	1.5 ± 0.3	−0.08 ± 0.1	**<0.01**	1.5 ± 0.3	−0.05 ± 0.1	**<0.01**	0.24
Triglycerides (mmol/L)	1.4 ± 0.5	0.06 ± 0.4	0.25	1.2 ± 0.6	0.16 ± 0.5	**0.02**	0.23

^1^ Mean ± SD, significant values *p* < 0.05 are presented in bold; ^2^ Change values represent the calculated delta between endline and baseline, after the four week intervention of either Kori-tofu or control. The *p*-values for the difference between the Kori-tofu and whey protein control intervention were calculated based on these delta values.

**Table 3 nutrients-15-00049-t003:** Glucose metabolism markers at baseline and change after 4 weeks Kori-tofu or control intervention ^1^.

	Kori-tofu Intervention			Control Intervention			Between Kori-tofu and Control
	Baseline	Change ^2^	*p*-Value	Baseline	Change ^2^	*p*-Value	*p*-Value
Glucose (mmol/L)	5.4 ± 0.6	0.04 ± 0.3	0.36	5.5 ± 0.5	−0.03 ± 0.4	0.63	0.39
Insulin (ulU/mL)	8.7 ± 5.2	0.56 ± 4.3	0.39	8.7 ± 4.5	0.1 ± 2.5	0.85	0.53
Fructosamine (μmol/L) ^3^	260 ± 20	−6.1 ± 11	**<0.01**	259 ± 17	−2.57 ± 13	0.22	0.17
Hemoglobin A1c (HbA1c) (mmol/mol)	37 ± 3.4	−0.42 ± 1.1	**0.02**	37 ± 14	0.11 ± 1.3	0.58	0.06
Homeostatic Model Assessment of Insulin Resistance (HOMA IR) score	2.2 ± 1.5	0.17 ± 1.1	0.32	2.2 ± 1.2	0.02 ± 0.7	0.98	0.39

^1^ Mean ± SD, significant values *p* < 0.05 are presented in bold; ^2^ Change values represent the calculated delta between end and baseline, after the four week intervention of either Kori-tofu or control. The *p*-values for the difference between the Kori-tofu and whey protein control intervention were calculated based on these delta values. ^3^ Fructosamine *N* = 44 instead of *N* = 45, since one sample was lost during shipment to the laboratory for fructosamine analysis, data for fructosamine of this subject were excluded in both interventions.

**Table 4 nutrients-15-00049-t004:** Blood pressure values at baseline and change after 4 weeks intervention with Kori-tofu or control intervention ^1^.

	Kori-tofu Intervention			Control Intervention			Between Kori-tofu and Control
	Baseline	Change ^2^	*p*-Value	Baseline	Change ^2^	*p*-Value	*p*-Value
Systolic blood pressure (mmHg)	122 ± 13	−3.1 ± 9.3	**0.03**	121 ± 15	−0.4 ± 8.1	0.72	0.18
Diastolic blood pressure (mmHg)	75 ± 8.4	−1.1 ± 5.1	0.15	74 ± 8.6	1.0 ± 5.7	0.24	0.06
Heart rate (BPM)	64 ± 10	1.7 ± 7.2	0.12	64 ± 8.0	3.8 ± 5.7	**<0.01**	0.14

^1^ Mean ± SD, significant values *p* < 0.05 are presented in bold; ^2^ Change values represent the calculated delta between end and baseline, after the four week intervention of either Kori-tofu or control. The *p*-values for the difference between the Kori-tofu and whey protein control intervention were calculated based on these delta values.

**Table 5 nutrients-15-00049-t005:** Leptin and adiponectin values at baseline and change after 4 weeks intervention with Kori-tofu or control intervention ^1^.

	Kori-tofu Intervention			Control Intervention			Between Kori-tofu and Control
	Baseline	Change ^2^	*p*-Value	Baseline	Change ^2^	*p*-Value	*p*-Value
Leptin (ng/mL)	3.1 ± 3.8	0.1 ± 0.9	0.67	3.1 ± 3.7	0.0 ± 0.8	0.76	0.56
Adiponectin (μg/mL)	106 ± 71	−14 ± 49	0.06	93 ± 52	8.9 ± 38.0	0.13	**0.02**

^1^ Mean ± SD, significant values *p* < 0.05 are presented in bold; ^2^ Change values represent the calculated delta between end and baseline, after the four week intervention of either Kori-tofu or control. The *p*-values for the difference between the Kori-tofu and whey protein control intervention were calculated based on these delta values.

## Data Availability

Data are available upon substantiated request from the corresponding author. Data are not publicly available since subjects did not sign consent for this.

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
