# Peer review of "Effect of 4-Week Consumption of Soy Kori-tofu on Cardiometabolic Health Markers: A Double-Blind Randomized Controlled Cross-Over Trial in Adults with Mildly Elevated Cholesterol Levels"

_nutrients, 2022, doi:10.3390/nu15010049_

Round 1

Reviewer 1 Report

The present work assessed the potential of Kori-tofu dietary supplementation during 4 weeks to improve cardiometabolic markers in adults with mild hypercholesterolemia and therefore potentially prevent cardiovascular diseases.

The clinical study was well-designed and methods were clearly described in the manuscript.

The results were in general clearly presented, however, I have a few comments and suggestions to make results clearer and more understandable:

- It is not clear if results presented in Table 2, 3, 4, and 5 include the data from the repeated 4 weeks (T=4 and T=12). If yes, this should be mentioned in the description.

- The standard deviation of change values in all tables are quite high indicating a high coefficient of variation, thus a high variation in participants response to treatment. Could the authors comment on this?

- I recommend to the authors to present the results in graphs too, this will give a clearer view of results variation after treatment. I would suggest to use box plot for example.

Minor format comments:

- Lines 205 and 241: It seems there's an error when including Table 1 and Table 3 to the text.

- the layout and format of tables should be checked.

- Line 273: "...adiponectin levels (p=0.03)" while it is 0.02 in Table 5.

Results were discussed thoroughly and limitations of the current study explained.

All conclusions were supported by the results.

Finally,  I strongly recommend to the authors to include graphs in the results section.

Reviewer 2 Report

Authors present a well-planned and properly executed trial. Nevertheless, one should clearly admit that the end-point was somehow disappointing. Some of the changes may have been statistically significant, but clearly lacked clinical significance. One change that could potentially means something was the >7% decline in LDL. This is not outstaning (may be comparable to some already tested plant-based supplements. Nevertheless, this may be interesting from clinical perspective. What I would suggest is to clearly state that most of those effects have severly limited clinical application.

Overall, this is an interesting study, definitely worth publishing. Thus, I suggest minor revisions.

Minor comments:
Multiple lines: "Error! References source not found."

Author Response

Please see the attachment.  The editor indicated that our manuscript should undergo extensive English revision. However, both reviewers indicated that English language and style are fine or minor spell check were required. We would like to hear if additional English check is required.
